

# Psychological flexibility, cognitive emotion regulation and mental health outcomes among patients with asthma in Pakistan

Samavia Hussain[1], Rabia Khawar[1], Rizwana Amin[2], Asma Hamdani[3] and Asma Majeed[4]

[1] Department of Applied Psychology, Faculty of Arts and Social Sciences, Government College University, Faisalabad, Faisalabad, Punjab, Pakistan
[2] Department of Professional Psychology Bahria University, Islamabad Campus Pakistan, Bahria University Islamabad Campus, Islamabad, Punjab, Pakistan
[3] Department of Applied Psychology, Faculty of Arts and Social Sciences, Government College University, Faisalabad, Faisalabad, Pakistan
[4] Department of Applied Psychology, Kinnaird College for Women, Lahore, Lahore, Punjab, Pakistan

Corresponding author
Rabia Khawar,
rabiakhawar@gcuf.edu.pk

## ABSTRACT

**Background/Objective:** Adults with asthma who experience difficulties in emotion regulation are prone to developing psychopathological symptoms that may affect their self-management activities and symptom control. The current research investigated the role of psychological flexibility and cognitive emotion regulation strategies in relation to mental health outcomes (psychological distress and quality of life) among patients with asthma in Pakistan.

**Method:** A sample of 200 adults, diagnosed with asthma (32% men, 68% women; $M_{age}$ = 42.32, $SD_{age}$ = 16.99), completed the acceptance and action questionnaire (AAQ-II) cognitive emotion regulation questionnaire (CERQ), depression, anxiety, stress scale-21 (DASS-21), asthma-related quality of life questionnaire (AQLQ) and a self-structured demographic sheet.

**Results:** Results of Pearson product moment correlation showed that most of the adaptive cognitive emotion regulation strategies (positive refocusing, refocus on planning, and positive reappraisal) were positively correlated with psychological flexibility and quality of life, whereas negatively correlated with psychological distress. All the maladaptive strategies of cognitive emotion regulation had a significant inverse relationship with psychological flexibility and quality of life, while positively correlated with psychological distress. Results of serial mediation analysis through PROCESS MACRO showed that catastrophising and anxiety fully mediated the relationship between psychological flexibility and asthma-related quality of life.

**Conclusion:** Evidence-based interventions should focus on developing psychological flexibility and identifying maladaptive patterns of cognitive emotion regulation strategies for improved mental health and quality of life outcomes for adults with asthma.

## INTRODUCTION

Asthma is a multi-factorial chronic respiratory illness, characterized by wheezing, coughing, shortness of breath and chest tightness (*World Health Organization, 2023*; *Global Initiative for Asthma, 2023*). The *World Health Organization (2023)* estimates that there were 383,000 casualties of asthma caused in 2015 around the globe. By 2025, the number of patients suffering from asthma is likely to increase up to 400 million (*Global Initiative for Asthma, 2023*). Pakistan is the 6th most populated country and more than twenty million individuals are facing asthmatic difficulties (*Chiesi Respiratory Diseases Center, 2016*).

People suffering from any chronic illness like diabetes, chronic pain, multiple sclerosis osteoporoses, and asthma *etc.*, reported a high level of psychological distress (*Mort & Philip, 2014*) and lower quality of life (*Trost et al., 2015*; *Preis et al., 2018*). Several researches indicated that almost in all three stages of asthma: onset, progression, and exacerbation (*Edwards et al., 2017*), people suffering from asthma had reported a low level of happiness, well-being, more psychological distress (anxiety, depression, & stress), and poorer quality of life in contrast to healthy controls (*Brumpton et al., 2013*; *Ampon et al., 2005*) and were at more risk of having asthma-related emergency room visits (*Ahmedani et al., 2013*). In a study conducted by *Pickles et al. (2018)*, asthma was found to be correlated with poor quality of life, exhaustion, restrictions in doing daily life activities effectively, adverse impacts on well-being, as well as with reduced productivity. Findings of studies conducted in Pakistan indicated that 15% of the asthmatic patients experienced depressive symptoms (*Motiani, Haidri & Rizvi, 2011*) especially, the women suffering from asthma (*Malik et al., 2017b*). Studies further showed that emotional problems related to physical illness could be considered a serious threat to a patient's quality of life (*Saleem, Mahmood & Naeem, 2016*). In Pakistan adults with asthma experience poor health-related quality of life but with the help of better medical facilities it can be improved (*Malik et al., 2017a*).

Certain factors could be associated with psychopathology and poorer quality of life including perception or acceptance of illness such as personality dispositions (*Monique Van De Ven & Engels, 2011*), psychological flexibility (*Guerrini Usubini et al., 2021*; *Bergman & Keitel, 2020*), and coping strategies (*Li et al., 2015*; *Kristofferzon, Engström & Nilsson, 2018*). Psychological flexibility is the most important variable of acceptance and commitment therapy (ACT) and emerged as a significant determinant of emotional well-being and behavioral value (*Bond, Lloyd & Guenole, 2013*). Psychological flexibility as an ability to build up a strong contact with the present moment in the presence of emotionally interfering thoughts (*Hayes et al., 2006*; *Moran, 2015*), has been significantly related to greater well-being (*Dawson & Golijani-Moghaddam, 2020*), a better quality of life and reduced psychological distress (*Wersebe et al., 2018*). Previous researches suggested that psychological flexibility proved to be a very useful intervention for a wide range of psychological disorders such as post-traumatic stress disorder, depression, anxiety, stress (*Powers, Vörding & Emmelkamp, 2009*), and for individuals suffering from any physical chronic illness (*Prevedini et al., 2011*). Psychological flexibility is significantly

associated with higher well-being among people with obesity and enables them to make interventions for healthy living (*Guerrini Usubini et al., 2021*). *Karimzadeh & Latifi (2015)* identified psychological flexibility as a strong predictor of improved quality of life and reduced negative emotional consequences of asthma.

Psychological flexibility also cope with stress resulting from chronic illness and enhanced resilience against chronic disease (*Gentili et al., 2019*; *Vowles et al., 2014*) might be helpful in the selection of adaptive coping strategies and related behaviors. Adaptive cognitive emotion regulation strategies like acceptance, putting into perspective, refocus on planning, positive refocusing, positive reappraisal (as cited in *Garnefski, Kraaij & Spinhoven (2001)*) and psychological flexibility had a positive impact on the quality of life of the people suffering from any chronic illness and enable people to cope effectively with their disease (*Li et al., 2015*; *Momeniarbat et al., 2017*). In contrast to psychological flexibility, inflexibility is particularly known as experiential avoidance (tendency to escape from challenging experiences, thoughts, feelings, and situations (*Hayes et al., 2006*) has found to be related to maladaptive coping and poorer quality of life, well-being and impaired psychological/emotional outcomes such as depression, anxiety, stress (*Fergus, Bardeen & Orcutt, 2013*; *Nielsen, Sayal & Townsend, 2018*). Maladaptive cognitive emotion regulation strategies including self-blame, rumination, catastrophising, and blaming others (as cited in *Garnefski, Kraaij & Spinhoven (2001)*) might also act as a central part in the perception of illness and also in the quality of life of the individuals suffering from chronic physical illness such cancer (*Postolica et al., 2017*). Catastrophising has been reportedly linked with psychological outcomes of physical illness such as poorer quality of life (*Sherwin, Leary & Henderson, 2017*). In an Iranian sample of asthma patients, it was revealed that catatstrophising proved to be a strong mediator between alexithymia and physical symptoms of asthma (*Ghorbani et al., 2017*). So, it can be seen that when a person suffers from any physiological disease, it will affect his/her quality of life and develop psychological distress among them. If the psychological flexibility is high then it will play an important role in maintaining a better quality of life and decreasing emotional distress. That is why it is very important to know how to cope with disease so that the quality of life will be maintained. Literature provides that no study was found, that focused on the association between cognitive-emotional regulation strategies and psychological flexibility with psychological distress and quality of life in patients living with asthma. Thus the current research intends (a) to check the association among cognitive emotion regulation coping techniques (adaptive, maladaptive), psychological/emotional flexibility, and mental health outcomes (psychological distress & asthma-related quality of life) among individuals suffering from asthma; (b) to explore the mediating effect of psychological distress and catastrophising on the association between psychological flexibility and health-related quality of life in patients with asthma.

## METHOD

### Participants and procedure

After getting ethical approval the procedure of data collection was started. This manuscript is the sub session of MS dissertation which was approved from board of studies at the

Department of Applied Psychology Government College University Faisalabad, Pakistan (reference no. Psy/206). Both men and women diagnosed with asthma since one year at least and receiving treatment at the time of research, were included. Patients with any other physiological co-morbidity (diabetes, heart failure, kidney failure *etc.*) and mental health problems were excluded. After seeking permission from the hospital authorities and clinics, written informed consent was obtained from patients diagnosed with asthma and their caregivers to fulfill the ethical requirement. Participants were instructed about the questionnaires and asked to complete cognitive emotion regulation questionnaire (CERQ), acceptance and action questionnaire-II (AAQ-II) depression, anxiety stress scale-21 (DASS-21), and asthma quality of life questionnaire (AQLQ). Authors have permission to use these instruments from copy right holders (permissions attached in Supplemental Files). Besides these questionnaires, a demographic information sheet was also used to record the essential demographics of the participants. At first 235 questionnaires were distributed from which 15 participants refused to participate in research; nine did not provide essential demographic information and 11 participants did not meet the inclusion criteria. Finally, a total sample of 200 individuals ($M_{age}$ = 42.32, $SD_{age}$ = 16.99) above 18 years, both men ($n$ = 63; $M_{age}$ = 48.84, $SD_{age}$ = 18.97) and women ($n$ = 137; $M_{age}$ = 39.32, $SD_{age}$ = 15.16) diagnosed with asthma since one year at least and receiving treatment at the time of research, were drawn from pulmonology unit of different public/private sector hospitals and clinics of Faisalabad (Pakistan). Purposive sampling technique and correlational research design were followed. It was made sure that all information was kept confidential and used only for research purposes.

## Instruments

1) **Cognitive emotional regulation questionnaire** (CERQ; *Garnefski, Kraaij & Spinhoven, 2002*). CERQ is a multi-dimensional self-report instrument containing 36 items, measures nine adaptive and maladaptive cognitive emotion regulation coping strategies on five points Likert scale. Five subscales measure adaptive strategies (acceptance, putting into perspective, positive refocusing, refocus on planning, & positive reappraisal) while, four subscales (catastrophising, rumination, self-blame & other blame) are categorized as maladaptive cognitive emotion regulation strategies. Each subscale has four items. Researches on cognitive emotion regulation questionnaire confirmed that each subscale showed good reliability coefficient (0.68 to 0.86); (*Garnefski, Kraaij & Spinhoven, 2002*). In current study Urdu version of CERQ (*Butt et al., 2016*) was used with adequate Chronbach alpha reliability (α = 0.71).

2) **Acceptance and action questionnaire** (AAQ-II; *Bond et al., 2011*). Acceptance and action questionnaire (AAQ) is a measure of psychological flexibility/experiential avoidance. It is seven points Likert scale high scores indicate higher levels of psychological inflexibility or experiential avoidance, while low scores indicate psychological flexibility (*Bond et al., 2011*). It has been widely used in health-related psychological studies and showed good psychometric properties (*Bond et al., 2011*;

*Zhang et al., 2014*; *Paladines-Costa et al., 2021*). For present study Urdu Version of AAQ-II consist of seven items (*Khawar & Aslam, 2018*) was used.

3) **Depression anxiety stress scale** (DASS–21; *Lovibond & Lovibond, 1995*). Depression anxiety stress scale was developed to measure the adverse impacts of three negative emotions including depression, anxiety and stress (*Lovibond & Lovibond, 1995*). The measure originally consists of 42 items on four point Likert scale. Its short-form called DASS-21 consists of 21 items; seven items for each subscale (*Lovibond & Lovibond, 1995*) having sound psychometric properties (*Pezirkianidis et al., 2018*; *Coker, Coker & Sanni, 2018*; *Bibi et al., 2020*). In current research, Urdu version of DASS-21 was used (*Farooqi & Habib, 2010*).

4) **Asthma quality of life questionnaire** (AQLQ; *Juniper et al., 1992*). Asthma quality of life questionnaire was used to assess the practical issues (psychological/emotional, physical, social and everyday work-related) experienced by the patients of asthma. There are 32 questions in AQLQ divided into four subscales including symptoms of asthma, limitations in daily life activities, emotional functions and environmental stimuli. The instrument provides an overall score for means of all items and mean scores for each domain (*Bateman et al., 2015*). Participants are instructed to complete this scale about how they have been experienced the last 14 days and to answer each of the 32 items on a seven point scale (7 = not impaired at all to 1 = severely impaired). It has excellent test-retest reliability ($r = 0.95$) (*Juniper et al., 1992*). For current research translated Urdu version of AQLQ with.87 Chronbach alpha reliability, was used.

5) **Demographic information sheet.** Demographic information (age, gender, education, number of family size, family system, monthly income, and duration of illness) of the adults with asthma was also obtained.

## RESULTS

The data were analyzed through the IBM statistical package for social sciences version 22 (SPSS, Inc., Chicago, IL, USA). The data were interpreted in the form of two sections, section I is labeled as "Descriptive Statistics" containing all the demographic information of the sample; while section II is named as "Inferential Statistics" consisting of correlation analysis, mediation, *etc.* A significance level of 0.05 was used for analysis.

Table 1 showed the demographic information of the sample such as; age, area of residence, marital status, education, duration of illness *etc.*

Table 2 shows the descriptive statistics and reliability coefficients for all the study measures and their subscales. Data are normally distributed as per criteria for social sciences (skewness & kurtosis ± 2) defined by *Gravetter & Wallnau (2014)*, and that is suitable for parametricstatistics.

Table 3 showed the results of Pearson Product Moment correlation between all the study variables. Both adaptive and maladaptive cognitive emotion regulation coping techniques were significantly correlated with psychological flexibility, psychological distress and domains of asthma related quality of life ($p < 0.001$, $p < 0.01$, $p < 0.05$). All adaptive cognitive emotion regulation strategies except acceptance and putting into

**Table 1 Demographic characteristics of study sample (N = 200).**

| | | Gender | | |
|---|---|---|---|---|
| Variables | Groups | Men<br>ƒ (%) | Women<br>ƒ (%) | Total<br>ƒ (%) |
| Age | 18–35 years | 19 (9.5) | 71 (35.5) | 90 (45) |
| | 36–55 years | 16 (8.0) | 42 (21.0) | 58 (29) |
| | 56–80 years | 28 (14.0) | 24 (12.0) | 52 (26) |
| | Total | 63 (31.5) | 137 (68.5) | 200 (100) |
| Marital status | Single | 6 (3.0) | 13 (6.5) | 19 (9.5) |
| | Married | 48 (24.0) | 101 (50.5) | 149 (74.5) |
| | Separated | 3 (1.5) | 7 (3.5) | 10 (5.0) |
| | Widowed/Widower | 6 (3.0) | 16 (8.0) | 22 (11.0) |
| | Total | 63 (31.5) | 137 (68.5) | 200 (100) |
| Area of residence | Rural | 24 (12.0) | 63 (30.0) | 84 (42) |
| | Urban | 39 (19.5) | 77 (38.5) | 116 (58) |
| | Total | 63 (31.5) | 137 (68.5) | 200 (100) |
| Family type | Nuclear | 38 (19.0) | 98 (49.0) | 136 (68) |
| | Joint/Extended | 25 (12.5) | 39 (19.5) | 64 (32) |
| | Total | 63 (31.5) | 137 (68.5) | 200 (100) |
| Monthly income | 10,000–50,000 | 50 (25.0) | 116 (58.0) | 116 (83) |
| | 51,000–100,000 | 12 (6.0) | 18 (9.0) | 30 (15) |
| | 101,000–150,000 | 1 (0.5) | 3 (1.5) | 4 (2) |
| | Total | 63 (31.5) | 137 (68.5) | 200 (100) |
| Education | Illiterate | 2 (1.0) | 20 (10) | 22 (11) |
| | Middle and below | 30 (15.0) | 53 (26.5) | 83 (41.5) |
| | Secondary/Higher Secondary | 19 (9.5) | 40 (20) | 59 (29.5) |
| | Graduation and above | 12 (6.0) | 24 (12.0) | 36 (18.0) |
| | Total | 63 (31.5) | 137 (68.5) | 200 (100) |
| Duration of illness | 1–5 years | 39 (19.5) | 91 (45.5) | 139 (65) |
| | 6–10 years | 14 (7.0) | 30 (15.0) | 44 (22) |
| | 11–15 years | 10 (5.0) | 16 (8.0) | 26 (13) |
| | Total | 63 (31.5) | 137 (68.5) | 200 (100) |

perspective were positively correlated with psychological flexibility and quality of life; while all maladaptive strategies and two adaptive strategies (putting into perspective & acceptance) had a significant negative association with psychological flexibility and quality of life.

Figure 1 showed the serial mediation analysis to identify the effect of psychological flexibility on quality of life through catastrophising and anxiety; while gender and duration of diagnosis of illness were taken as covariates and none of them were found to be significant. Serial mediation analysis (Model 6) was performed by estimating 5,000 bootstrap sample through PROCESS MACRO (*Hayes, 2022*). Findings indicated that total effect of psychological flexibility on quality of life was significant (b = 0.017,

**Table 2 Descriptives statistics and alpha level of all the study measures and their subscales.**

| Variables | M | SD | Skewness | Kurtosis | α |
|---|---|---|---|---|---|
| AAQ-II | 22.60 | 6.97 | 0.68 | 0.12 | 0.89 |
| CERQ | | | | | 0.71 |
| Positive refocusing | 11.41 | 3.42 | 0.52 | −0.48 | 0.80 |
| Acceptance | 12.68 | 2.53 | −0.11 | −0.33 | 0.68 |
| Positive reappraisal | 11.25 | 3.09 | 0.66 | −0.23 | 0.76 |
| Refocus on planning | 13.63 | 2.65 | 0.12 | −0.07 | 0.73 |
| Putting into perspective | 12.91 | 2.08 | −0.08 | 1.15 | 0.45 |
| Self–blame | 11.68 | 2.35 | −0.29 | −0.27 | 0.57 |
| Rumination | 13.09 | 1.95 | −0.81 | 1.67 | 0.49 |
| Catastrophising | 14.18 | 3.24 | −0.27 | −0.35 | 0.81 |
| Other blame | 11.46 | 2.41 | −0.12 | 0.92 | 0.69 |
| DASS-21 total | 68.66 | 14.63 | 0.12 | 0.00 | 0.82 |
| Depression | 22.05 | 6.63 | −0.21 | 0.12 | |
| Anxiety | 22.85 | 6.02 | 0.14 | −0.11 | |
| Stress | 23.76 | 4.84 | 0.23 | 0.60 | |
| AQLQ total | 3.07 | 0.52 | −0.70 | 0.34 | 0.87 |
| Activity limitations | 3.53 | 0.68 | −0.35 | −0.16 | |
| Symptoms | 3.22 | 0.61 | −0.71 | 0.89 | |
| Emotional function | 2.76 | 0.73 | 0.17 | 0.24 | |
| Environmental stimuli | 3.41 | 0.73 | 0.08 | 0.59 | |

$p < 0.01$; 95% CI [0.007–0.027]) while direct effect was non-significant ($b = −0.003$, 95% CI [−0.014-0.008]). On the other hand, indirect effect by assuming 95% CI were as follows: psychological flexibility catastrophising quality of life ($b = 0.124$, 95% CI [0.022–0.232]), psychological flexibility anxiety quality of life ($b = 0.023$, 95% CI [−0.039-0.087]), psychological flexibility catastrophising anxiety quality of life ($b = 0.120$, 95% CI [0.068–0.184]) (see Table S1). Total indirect effect of psychological flexibility on quality of life was 28%. The overall findings indicated that catastrophising and anxiety serially, fully mediated the relationship between psychological flexibility and quality of life. Findings showed that an increase in psychological flexibility predicted the low level of catastrophising while by increasing catastrophising an increase in anxiety was shown. Moreover, an increase in anxiety predicted a decrease in quality of life among patients with asthma.

Table S1 showed the total, direct and indirect effects of psychological flexibility on quality of life.

## DISCUSSION

Although the current study has provided significant contributions to the field of health psychology, but the following limitations were observed: in the present research, the sample was obtained from few hospitals of Faisalabad. In future the data would be collected

Hussain et al. (2023), *PeerJ*, DOI 10.7717/peerj.15506

**Table 3 Inter-correlation among psychological flexibility, adaptive and maladaptive cognitive emotion regulation strategies, psychological distress and quality of life ($N = 200$).**

| Variables | 1 | 2 | 3 | 4 | 5 | 6 | 7 | 8 | 9 | 10 | 11 | 12 | 13 | 14 | 15 | 16 | 17 | 18 |
|---|---|---|---|---|---|---|---|---|---|---|---|---|---|---|---|---|---|---|
| 1. Psychological flexibility | 1 | | | | | | | | | | | | | | | | | |
| 2. Acceptance | −0.44*** | 1 | | | | | | | | | | | | | | | | |
| 3. Positive refocusing | 0.44*** | −0.54*** | 1 | | | | | | | | | | | | | | | |
| 4. Refocus on planning | 0.20** | −0.32*** | 0.69*** | 1 | | | | | | | | | | | | | | |
| 5. Positive reappraisal | 0.41*** | −0.44*** | 0.82*** | 0.68*** | 1 | | | | | | | | | | | | | |
| 6. Putting into perspective | −0.28*** | 0.16* | 0.048 | 0.15* | 0.12 | 1 | | | | | | | | | | | | |
| 7. Self-blame | −0.47*** | 0.51*** | −0.42*** | −0.18* | −0.38*** | 0.25*** | 1 | | | | | | | | | | | |
| 8. Rumination | −0.50*** | 0.31*** | −0.12 | 0.15* | −0.15* | 0.30*** | 0.44*** | 1 | | | | | | | | | | |
| 9. Catastrophising | −0.59*** | 0.63*** | −0.64*** | −0.35*** | −0.56*** | 0.28*** | 0.60*** | 0.49*** | 1 | | | | | | | | | |
| 10. Other blame | −0.46*** | 0.42*** | −0.42*** | −0.34*** | −0.37*** | 0.27*** | 0.45*** | 0.34*** | 0.58*** | 1 | | | | | | | | |
| 11. Activity limitations AQLQ | 0.17* | −0.19** | 0.35*** | 0.27*** | 0.34*** | −0.06 | −0.21 | −0.10 | −0.32*** | −0.30*** | 1 | | | | | | | |
| 12. Symptoms AQLQ | 0.24** | −0.29*** | 0.36*** | 0.33*** | 0.33*** | −0.22** | −0.22 | −0.12 | −0.40*** | −0.30*** | 0.71*** | 1 | | | | | | |
| 13. Emotional function AQLQ | 0.19** | −0.25*** | 0.40*** | 0.36*** | 0.33*** | −0.13 | −0.24 | −0.07 | −0.36*** | −0.32*** | 0.61*** | 0.75*** | 1 | | | | | |
| 14. Environmental stimuli AQLQ | 0.08 | −0.10 | 0.04 | 0.11 | 0.02 | −0.02 | −0.09 | −0.02 | −0.03 | −0.22** | −0.10 | 0.10 | 0.11 | 1 | | | | |
| 15. AQLQ total | 0.23** | −0.28*** | 0.40*** | 0.35*** | 0.36*** | −0.17* | −0.25*** | −0.11 | −0.40*** | −0.36*** | 0.17* | 0.24** | 0.19** | 0.08 | 1 | | | |
| 16. Depression | −0.49*** | 0.57*** | −0.55*** | −0.39*** | −0.51*** | 0.15* | 0.54*** | 0.30*** | 0.60*** | 0.49*** | −0.30*** | −0.40*** | −0.38*** | −0.11 | −0.49*** | 1 | | |
| 17. Anxiety | −0.34*** | 0.43*** | −0.51*** | −0.44*** | −0.47*** | 0.11 | 0.36*** | 0.12 | 0.52*** | 0.34*** | −0.44*** | −0.49*** | −0.43*** | −0.01 | −0.35*** | 0.65*** | 1 | |
| 18. Stress | −0.48*** | 0.32*** | −0.41*** | −0.28*** | −0.40*** | 0.22** | 0.47*** | 0.31*** | 0.52*** | 0.46*** | −0.25*** | −0.31*** | −0.29*** | −0.12 | −0.48*** | 0.55*** | 0.39** | 1 |

**Notes:**
*** $p < 0.001$.
** $p < 0.01$.
* $p < 0.05$.
AQLQ, Asthma related quality of life questionnaire.

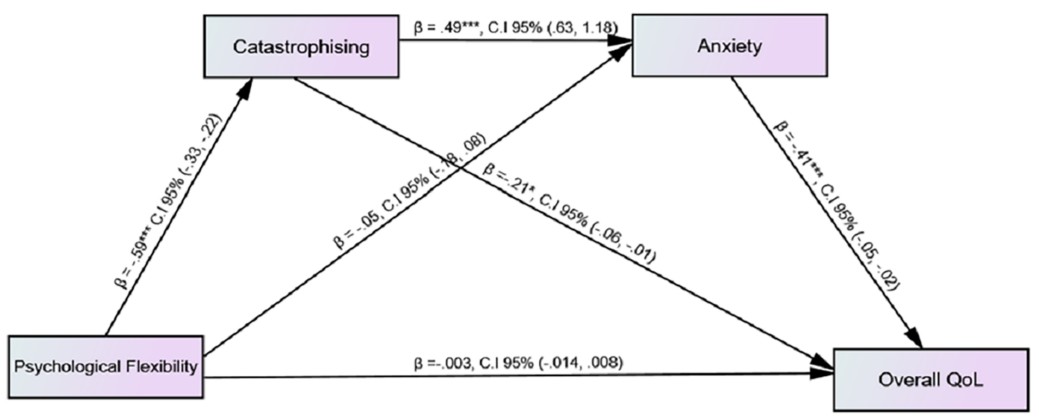

**Figure 1 Mediating effect of catastrophising and anxiety on the association among psychological flexibility and quality of life.**

from different cities in exchange for generalizability. There is no control group in this study that is a major limitation. A control group would yield the more degree of evidence for results. The sample size was restricted due to the shortage of time and reduces the chances of generalization of current research. Some participants showed hesitation in giving their personal information. A qualitative analysis should be incorporated due to the subjectivity of many factors involved. The findings of current research would be implicated in health psychology and counseling for patients with asthma.

The findings of current study showed that psychologically flexibility is significantly correlated with lower levels of psychological distress (including depression, anxiety and stress on DASS 21) and health-related quality of life (activity limitation, physical symptoms and emotional function on AQLQ). This association was however stronger for depression and quality of life based on physical symptoms. Previous studies illustrated that psychological flexibility has been constantly related to lower psychological distress and better emotional or psychological well-being among individuals suffering from chronic illness (*Tyndall et al., 2020*; *Wersebe et al., 2018*). Psychological flexibility could also be helpful in better disease management and leads toward a better quality of life among patients suffering from chronic respiratory disease in Hong Kong (*Cheung & Mak, 2016*). The underline mechanism of this positive association was further explored in terms of cognitive emotion regulation strategies.

Positive link between psychological flexibility, adaptive cognitive emotion regulation strategies, lower level of distress and higher quality of life found in present study. Findings showed that psychological flexibility and adaptive cognitive emotion regulation strategies are significantly positively correlated with each other and psychological flexibility serves as a buffer against psychological distress. Literature proved that psychological flexibility could reduce the negative effects of current life stressors on mental health and well-being (*Gloster, Meyer & Lieb, 2017*; *Fonseca et al., 2020*). Current study indicated that maladaptive cognitive emotion regulation coping techniques and psychological flexibility were inversely correlated with each other. AAQ-II in terms of high scores is considered as an instrument for measuring experiential avoidance (*Bond et al., 2011*). In our study

experiential avoidance/inflexibility had been particularly linked with catastrophising and rumination. Both of these cognitive emotion regulation strategies are not only considered maladaptive coping (*Garnefski & Kraaij, 2018*) but also have a significant role in general cognitive distortions (*Mahaffey et al., 2013*). *Rueda & Valls (2020)* revealed that psychological flexibility had an indirect impact on distress and well-being mediated through avoidant coping techniques which indicated that by using avoidant/maladaptive coping strategies the effect of psychological flexibility turns into inflexibility and had adverse impacts on quality of life.

Cognitive emotion regulation strategies have been significantly linked with psychological problems such as depression, anxiety, stress (*Garnefski & Kraaij, 2006*, *2007*; *Gross & Jazaieri, 2014*; *Liu & Thompson, 2017*) and quality of life (*Li et al., 2015*, *Kanwa & Iftikhar, 2019*). Our results indicated that adaptive cognitive emotion regulation strategies especially positive refocusing, refocus on planning, and positive reappraisal were significantly negatively associated with psychological distress (depression, anxiety, stress) and positively linked with asthma related quality of life. Current research also highlighted the significant positive correlation between maladaptive cognitive emotion regulation coping (rumination, catastrophising, self-blame, & other blame) and psychological distress. Literature provides significant verdicts for these findings that more use of adaptive strategies such as positive reappraisal, refocus on planning and positive refocusing was negatively associated with anxiety and depression (*Min et al., 2013*; *Barberis et al., 2017*) while positively linked with better quality of life (*Li et al., 2015*) among patients with chronic illness. Rumination and self-blaming were found to be directly associated with depression; catastrophising and other-blame were exclusively related to more anxious signs and symptoms (*Garnefski & Kraaij, 2018*). Literature proved that the person who uses maladaptive cognitive coping experienced more psychological problems like depression, anxiety, stress; and poor emotional well-being (*Bahrami et al., 2017*; *Garnefski, Hossain & Kraaij, 2017*). In a study conducted in Serbian breast cancer patients' support our findings that catastrophising and rumination are significant maladaptive cognitive coping strategies which would be helpful in changing the intensity of psychological distress and reflected in person's health related quality of life (*Kovač et al., 2020*).

Further, it was revealed that an increase in acceptance was highly significantly linked with an increase in depression, anxiety, stress and poorer quality of life, which is consistent with previous literature (*Domaradzka & Fajkowska, 2018*; *Dubey, Podder & Pandey, 2020*; *Manju, 2017*). Although acceptance is an adaptive strategy, few studies suggested that acceptance should be treated as maladaptive (*Martin & Dahlen, 2005*, *Tuna & Bozo, 2012*). As in acceptance subscale, some items ("I think I cannot change anything", "I must learn to live with this situation", *etc.*) appeared as maladaptive and may provide a sense of hopelessness or helplessness for current research. Moreover, our results also showed that increased use of putting into perspective as an adaptive coping strategy was significantly related to high level of depression and stress and poor quality of life. Previous research also showed negative relationship between this strategy and quality of life (*Li et al., 2015*). It may cause due to an uncertain situation of disease as when individuals suffer from chronic illness it was very difficult for them to focus on thinking and planning positive

events or situations in their lives and it becomes more stressful. Hence, it was revealed by our findings that the more one fails to put less emphasis on stressful things or situations and tries to accept his/her physical illness, the more it will be depressing, stressful and helpless for him/her; the more one tries to focus on positive and pleasant things in life, engaging in thinking positive aspects of the situation instead of the negative situation, and making plans about deciding what steps should be taken to overcome the stress, the more sense of worth he/she will develop.

Results of serial mediation analysis using PROCESS MACRO (*Hayes, 2022*) revealed catastrophising (maladaptive cognitive emotion regulation strategy) and anxiety mediated the link between psychological flexibility and quality of life. When we introduce catastrophising as mediator the effect of psychological flexibility on quality of life was diminished because the higher catastrophising lowers the impact of psychological flexibility/higher experiential avoidance and leads towards the poorer quality of life. Psychological flexibility as a personality trait has shown greater openness in accepting life stressors and carrying valued living. In this way, psychological flexibility is positively associated with greater individual well-being and quality of life (*Berghoff et al., 2014*; *Ramaci et al., 2019*). Psychological flexibility also minimized emotional distress and enhanced better psychological health of patients with asthma (*Karimzadeh & Latifi, 2015*). It was found in previous researches that coping and psychological distress (anxiety) was significantly mediated between personality traits and health-related quality of life (*Pereira-Morales et al., 2018*). Literature supports our findings that quality of life among patients suffering from asthma was mediated through cognitive coping techniques (*Van Lieshout & MacQueen, 2012*). Previous researches provide strong verdicts that among cognitive emotion regulation strategies just catastrophising, plays a mediating role among alexithymia subscales and physical symptoms among asthmatics patients, in this way catastrophizing intensifies the asthmatic symptoms and leads toward greater psychological distress and poorer quality of life (*Rieffe et al., 2010*). In another study, it was found that catastrophising and psychological distress significantly mediated the link between pain-related fear and quality of life among people with chronic low back pain (*Marshall, Schabrun & Knox, 2017*). Catastrophising is directly related to psychological distress and in this way more emotional issues are related to this coping style.

## CONCLUSION

The present study provides one of the first looks at the role of different cognitive emotion regulation strategies (adaptive, maladaptive) and gives an insight into the frequency of these strategies, psychological flexibility, and their impacts on quality of life. This study will prove to be a significant addition in literature not only for health psychology as well as for better asthma management in medical sciences and provide strong foundations for further researches regarding this issue.

## ACKNOWLEDGEMENTS

We acknowledge all the participants who took part in research.

### Funding
The authors received no funding for this work.

### Competing Interests
The authors declare that they have no competing interests.

### Author Contributions
- Samavia Hussain conceived and designed the experiments, performed the experiments, analyzed the data, prepared figures and/or tables, and approved the final draft.
- Rabia Khawar conceived and designed the experiments, performed the experiments, analyzed the data, authored or reviewed drafts of the article, and approved the final draft.
- Rizwana Amin performed the experiments, analyzed the data, authored or reviewed drafts of the article, and approved the final draft.
- Asma Hamdani conceived and designed the experiments, prepared figures and/or tables, and approved the final draft.
- Asma Majeed performed the experiments, analyzed the data, prepared figures and/or tables, authored or reviewed drafts of the article, and approved the final draft.

### Ethics
The following information was supplied relating to ethical approvals (*i.e.*, approving body and any reference numbers):

Institutional Review Board Government College University Faisalabad. The Board of Studies Department of Applied Psychology, Government College University Faisalabad Pakistan approved the study (Reference No. Psy/206).

### Data Availability
Raw data is available in the Supplemental Files.

This file contain raw data of all variables included in study.

### Supplemental Information
Supplemental information for this article can be found online at http://dx.doi.org/10.7717/peerj.15506#supplemental-information.

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
