# Peer review of "Psychological flexibility, cognitive emotion regulation and mental health outcomes among patients with asthma in Pakistan"

_PeerJ, doi:10.7717/peerj.15506_

## Round 0.1 · original submission · Major Revisions

I have now received the reviewers' comments on your manuscript. They have suggested major revisions to your manuscript. Therefore, I invite you to respond to the reviewers' comments and revise your manuscript.

·

Basic reporting

The manuscript reports the analysis of the role of psychological flexibility and emotion regulation among people suffering from asthma in Pakistan.
The topic is interesting and I think the methods are clear altough I have some major concern for the authors looking to improve their paper.

The whole manuscript needs significant editing for language and writing quality. Examples in particular where the language needs to be improved, as it results difficult to be understood, include lines: 83-84 (and please refer to people with obesity in a more polite way than just "obese"), 88-89, 142-143, 147, 168-170, 193, 203-204, 227-229, 246-247, 250 and many more in the "Discussion" paragraph. Authors should revise the language to improve readability, please review the sequence of tenses in the "Instruments" section also.

The study fails to address how the findings relate to previous research in the area. The authors should rewrite their "Introduction" and "Discussion" to reference some relevant literature. I just cite some papers beetween the most cited in similar literature, but a better comprehensive research should be done:
- Bosley, C. M., Fosbury, J. A., & Cochrane, G. M. (1995). The psychological factors associated with poor compliance with treatment in asthma. European Respiratory Journal, 8(6), 899-904.
- Adams, R. J., Wilson, D. H., Taylor, A. W., Daly, A., d’Espaignet, E. T., Dal Grande, E., & Ruffin, R. E. (2004). Psychological factors and asthma quality of life: a population based study. Thorax, 59(11), 930-935.
- Baiardini, I., Sicuro, F., Balbi, F., Canonica, G. W., & Braido, F. (2015). Psychological aspects in asthma: do psychological factors affect asthma management?. Asthma research and practice, 1(1), 1-6.
- Stanescu, S., Kirby, S. E., Thomas, M., Yardley, L., & Ainsworth, B. (2019). A systematic review of psychological, physical health factors, and quality of life in adult asthma. NPJ primary care respiratory medicine, 29(1), 1-11.
Also the "Participants & Procedure" section should be rewritten in a more clear way and following more rigorous criteria and a conceptual change. Examples: information in line 122 is totally irrelevant; in line 127 are cited inclusion criteria never discussed in the methodology, it is possible to undestand that there are some reading the following lines 128-131 but this is not clear at all.

Limitations are discussed in the conclusion. Information about the limitations of a reaserch are generally placed at the beginning of the discussion section so the reader knows and understands the limitations before reading the rest of analysis of the findings. Please review carefully the "Conclusion" paragraph after reviewing the rest of the manuscript.

Minor concerns:
Please review affiliation numbers in the title section (lines 5 and 7 are duplicates)

Abstract should not contain blibliographic reference (lines 26, 27, 28, 29), please move them to the methods section.

Line 238: it's clear that the abbreviation QoL states for quality of life, but it's the first time you are using it in the paper without specifying.

Lines 334 and 338: shortage of time could be a limitation for the due date of a dissertation, not for sampling research participants...

Experimental design

no comment

Validity of the findings

no comment

Reviewer 2 ·

Basic reporting

Some sentences are confusing. See the supplementary comments for details.

Experimental design

The theoretical meaning of the research is not clear, and relevant explanations can be supplemented in the introduction part.

Validity of the findings

The title, conclusions and data analysis results of the paper do not match well

Additional comments

Q1: Can add what is the theoretical basis for the establishment of the research model in the introduction part?

Q2:line118-119,to explore the mediating effect of psychological distress and catastrophising on the association between psychological flexibility and health-related quality of life in patients with asthma.The title of the paper is ‘Psychological Flexibility, Cognitive Emotion Regulation and Mental Health Outcomes’,but the paper only explore the mediating effect of psychological distress and catastrophising on the association between psychological flexibility and health-related quality of life. The paper also does not give a special explanation why only catastrophising was studied. In addition, the order of psychological distress and catastrophising is inconsistent with that in the model.

Q3:line156-163, How many items does Acceptance and Action Questionnaire have?


Q4: Why is there no reliability coefficient of subscales like Depression in Table 3? Please add.

Q5:line 215-216, while gender and duration of diagnosis of illness were taken as covariates. Why did you choose these two demographic variables as control variables instead of other demographic variables? Please provide relevant reasons.

Q6: line218-219 and table4 ‘total effect of psychological flexibility on quality of life was significant, b = .017. Isn't the total effect equal to the sum of the direct effect (not significant) and the significant indirect effect? According to the description, I think the total effect is equal to 0.124+0.120.

Q7:Why only anxiety was studied in the model, while depression and stress were also measured when measuring. Especially mentioned in the discussion ‘This association was however stronger for depression and QoL based on physical symptoms(line 239)’. So why not study depression? Please give a description of the reasons.

Q8:line259-262. Rueda and Valls (2020) revealed that psychological flexibility had an indirect impact on distress and well-being mediated through avoidant coping techniques which indicated that by using avoidant/maladaptive coping strategies the effect of psychological flexibility turns into inflexibility and had adverse impacts on quality of life. I'm a little confused about the meaning of the second half of this sentence. Is it accurate?

Q9: line 263-283,The article discussed many literatures to illustrate ‘Cognitive emotion regulation strategies have been significantly linked with psychological problems such as depression, anxiety, stress and quality of life’. If there were so many relevant researches, where is the innovation of this research? This needs to be supplemented.

Q10. line307-308, because the higher catastrophising lowers the psychological flexibility and leads towards the poorer quality of life. catastrophising lowers the psychological flexibility? So does catastrophising affect psychological flexibility?In the front of this paper, it is said that psychological flexibility affect catastrophizing (see Figure 1). So, who is the cause variable and who is the result variable?

Q11. line311-312,Psychological flexibility also minimized emotional distress and psychological health of patients with asthma (Karimzadeh & Latifi, 2015). Psychological flexibility also minimized psychological health of patients with asthma?

Q12. line316, Literature supports our findings that quality of life among patients suffering from asthma was mediated through cognitive coping techniques (Van Lieshout & MacQueen, 2012). I feel this expression is strange, what was mediated through cognitive coping techniques?

Q13. line326-328, The present study provides one of the first looks at the role of different cognitive emotion regulation strategies (adaptive, maladaptive) and gives an insight into the frequency of these strategies, psychological flexibility, and their impacts on quality of life. However, this paper only makes a correlation analysis of the relationship between these variables, and the model tests only analyzed the role of catastrophizing, not all cognitive emotion regulation strategies (adaptive, maladaptive).

---

## Round 0.2 · accepted · Accept

I would like to thank the authors for addressing all reviewers' comments.

·

Basic reporting

no comment

Experimental design

no comment

Validity of the findings

no comment

Additional comments

Thank you for addressing all of my comments. Your commitment is truly appreciated.

Reviewer 2 ·

Basic reporting

no comment

Experimental design

no comment

Validity of the findings

Please consider expressing the findings of this study in more detail and clarity in the conclusion section.

Additional comments

no comment